# Changes in the Physical Properties of Calcium Alginate Gel Beads under a Wide Range of Gelation Temperature Conditions

**DOI:** 10.3390/foods9020180

**Published:** 2020-02-12

**Authors:** Chungeun Jeong, Seonghui Kim, Chanmin Lee, Suengmok Cho, Seon-Bong Kim

**Affiliations:** Department of Food Science and Technology, Institute of Seafood Science, Pukyong National University, Busan 48513, Korea; jeongpknu@gmail.com (C.J.); shkim.pknu@gmail.com (S.K.); 3edbn3@gmail.com (C.L.); scho@pknu.ac.kr (S.C.)

**Keywords:** calcium alginate gel, gelation temperature, physical property, response surface methodology, preparation condition

## Abstract

Until now, most studies using calcium alginate gel (CAG) have been conducted primarily at room temperature (20 °C) without considering gelation temperature. Moreover, the effects of gelation temperature on the physical properties of CAG beads have not been studied in detail. We aimed to study the effect of gelation temperature on the physical properties (diameter, sphericity, and rupture strength) of CAG beads. Response surface methodology was used in this study. The independent variables were sodium alginate concentration (X_1_, 1.2–3.6%, *w*/*v*), calcium lactate concentration (X_2_, 0.5−4.5%, *w*/*v*), gelation temperature (X_3_, 5–85 °C), and gelation time (X_4_, 6–30 min). Diameter (Y_1_, mm), sphericity (Y_2_, %), and rupture strength (Y_3_, kPa) were selected as the dependent variables. A decrease in gelation temperature increased the diameter, sphericity and rupture strength of the CAG beads. Additionally, the CAG beads prepared at 5 °C exhibited the highest rupture strength (3976 kPa), lowest calcium content (1.670 mg/g wet), and a regular internal structure. These results indicate that decreasing the gelation temperature slows the calcium diffusion rate in CAG beads, yielding a more regular internal structure and increasing the rupture strength of the beads.

## 1. Introduction

Calcium alginate gels (CAGs) have been used widely in various fields of biotechnology, including the food, medicine, and pharmaceutical industries, due to their biocompatibility, low toxicity, easy gel formation, and low price [1,2]. CAG formation is associated with the characteristic structure of alginate, a linear copolymer of 1,4-linked β-D-mannuronic acid and α-L-guluronic acid in which homopolymeric stretches of guluronic acid residues cooperatively bind calcium ions to form a three-dimensional gel structure, known as the egg-box model [3,4].

Generally, CAGs are prepared as a variety of beads or capsules [5,6]; however, CAG beads are preferred over capsules as their preparation method is simpler. In the food industry, CAG beads are used to prepare imitation foods (particularly artificial fish roe) and encapsulate bioactive components (including antimicrobials, antioxidants, nutraceuticals, and flavors). The physical properties of CAG beads are very important [7,8], including their diameter, sphericity, and rupture strength which are key factors for the development of imitation foods because they are responsible for the appearance and texture of the products. Moreover, during encapsulation, the diameter and strength of the CAG beads are important to control the diffusion of bioactive components and the crushing of capsules owing to the contractile force of the stomach and small intestine, respectively [9,10].

To date, in most previous studies, CAGs have been prepared at room temperature (approximately 20 °C) [5,11,12,13]. In addition, there is limited information regarding the effects of gelation temperature on the physical properties of CAGs. Yamagiwa et al. [14] investigated the effect of gelation temperature (5–55 °C) on compression strength. Although gelation temperature might be a significant factor in CAG preparation, the effects of a wide range of gelation temperatures on various physical properties of CAGs have not been studied yet. Therefore, in this study, we investigated the physical properties of CAG beads prepared under a wide range of gelation temperatures (5−85 °C). The diameter, sphericity, and rupture strength were considered physical properties of CAG beads.

Response surface methodology (RSM) is an effective mathematical and statistical method to monitor or optimize processes when several factors and interactions influence the process response [15]. Thus, RSM is extensively used during the monitoring and optimization of food processing [16]. Central composite design (CCD) is one of the most popular experimental designs of RSM to obtain a quadratic polynomial model describing the experimental system [17,18]. In this study, RSM and CCD were used to monitor the effects of gelation temperature along with other factors including sodium alginate and calcium lactate concentration and gelation time. The independent variables (factors) used this study were sodium alginate concentration (X_1_, 1.2−3.6%, *w*/*v*), calcium lactate concentration (X_2_, 0.5−4.5%, *w*/*v*), gelation temperature (X_3_, 5–85 °C), and gelation time (X_4_, 6–30 min), while diameter (Y_1_, mm), sphericity (Y_2_, %), and rupture strength (Y_3_, kPa) were selected as the dependent variables. In addition, to better understand the relationship between rupture strength of the CAG bead and gelation temperature, calcium and sodium ion contents and microstructures of CAG beads were investigated.

## 2. Materials and Methods

### 2.1. Materials

Sodium alginate (molecular weight: 220,000) and calcium lactate were purchased from Junsei Chemical Co., Ltd. (Tokyo, Japan) and Daejung Chemicals and Metals Co., Ltd. (Gyeonggi, Korea), respectively. Standard solutions for measuring sodium and calcium ion content were obtained from AccuStandard (1000 µg/mL in 2−5% nitric acid; Sodium ICP Standard, New Haven, CT, USA) and PerkinElmer (100 µg/mL in 5% HNO_3_; Quality Control Standard-21 Elements, MA, USA), respectively. All other chemicals and reagents used were of analytical grade.

### 2.2. Calcium Alginate Gel (CAG) Bead Preparation Method

CAG beads were prepared according to the methods of Ha et al. [7], with some modifications (Figure 1). Sodium alginate solution was dropped into calcium lactate solution at a flow rate of 0.03 mL/sec through a single nozzle (19G, inner diameter: 0.80 mm, outer diameter: 1.10 mm) using a peristaltic pump (SMP-23, Eyela, Tokyo, Japan). The temperatures of the sodium alginate and calcium lactate solutions were controlled by a heating and cooling bath circulator (RBC-22, LABHOUSE, Gyeonggi, Korea) and a heating agitator, respectively. The calcium lactate solution (250 mL) was agitated at a rate of 300 rpm in the reactor (500 mL). The drop distance from the single nozzle tip to the surface of the calcium lactate solution was 8 cm; therefore, the sodium alginate solution was affected by the room temperature (20 °C) while dripping into the calcium lactate solution surface, although it was difficult to measure this change accurately. We minimized the effect of room temperature by adjusting the outlet temperature of the sodium alginate solution from the nozzle to the gelation temperature. Only CAG beads prepared at 5 °C had been made in a cold chamber to maintain the temperature of the calcium lactate solution during the gelation time. All the prepared CAG beads were thoroughly washed with distilled water and used for analysis.

### 2.3. Diameter and Sphericity Measurement

To determine the diameter and sphericity of the CAG beads, we measured the shortest and longest diameter of ten randomly selected CAG beads with an image analyzer (i-Solution^TM^ 9.1, IMT i-Solution Inc., Daejeon, Korea) coupled to a stereoscopic microscope (125× magnification; SZX16, Olympus, Tokyo, Japan). The diameter (mm) of the CAG beads was calculated by averaging the shortest and longest diameters, while the sphericity (%) of the CAG beads was calculated as the percentage ratio of the shortest and longest diameters.

### 2.4. Rupture Strength Measurement

Rupture strength (kPa) is the maximum load applied to the sample area by a plunger when the sample is ruptured and permanently deformed. The rupture strength of the CAG beads (*n* = 10) was measured using a rheometer (Compac-100, Sun Scientific Co., Ltd., Tokyo, Japan) under the following conditions: MODE 4; adapter type, cylindrical plunger (diameter: 25 mm); compression speed, 60 mm/min; correction, 0.2 N; and load-cell, 0.1 kN.

### 2.5. Experimental Design and Statistical Analysis

CCD was used to monitor the effects of different preparation conditions on the physical properties of CAG beads. The CCD matrix was composed to 2^4^ factorial points, 2^3^ axial points (α = 2), and three replicates of the center point. The independent variables were sodium alginate concentration (X_1_, %, *w*/*v*), calcium lactate concentration (X_2_, %, *w*/*v*), gelation temperature (X_3_, °C), and gelation time (X_4_, min). The ranges of the independent variables and their levels are presented in Table 1. Diameter (Y_1_, mm), sphericity (Y_2_, %), and rupture strength (Y_3_, kPa) were chosen as the dependent variables and the run order of the experiment was randomized to minimize the effect of unexpected variables. The experimental data were analyzed using the response surface regression procedure in Minitab statistical software (Version 16, Minitab Inc., State College, PA, USA) to fit the following generalized quadratic polynomial model Equation (1):(1)$$Y=\beta_{0}+\overset{4}{\underset{i=1}{\sum }}\beta_{i}X_{i}+\overset{4}{\underset{i=1}{\sum }}\beta_{ii}X_{i}^{2}+\overset{3}{\underset{i=1}{\sum }}\overset{4}{\underset{j=i+1}{\sum }}\beta_{ij}X_{i}X_{j}$$
where Y is the predicted dependent variable, β_0_ is a constant, and β*_i_*, β*_ii_*, and β*_ij_* are linear, quadratic, and interaction regression coefficients, respectively. X*_i_* and X*_j_* are coded values of the independent variables. Three-dimensional response surface plots were produced from the fitted response surface model equations using Maple software (Maple 7, Waterloo Maple Inc., Waterloo, ON, Canada).

### 2.6. Moisture Content

The moisture content of CAG beads was determined using a digital moisture analyzer (MX-50, A & D, Tokyo, Japan). The temperature of the infrared drying chamber of the machine was set at 100 °C. Measurements were repeated 10 times and 50 CAG beads were used at a time. Fifty CAG beads were dried until there was no weight change of 0.1% in 1 min (based on wet weight). The moisture content was the percentage difference between the wet and dry weight divided by the wet weight.

### 2.7. Calcium and Sodium Ion Content

Inductively coupled plasma optical emission spectroscopy (ICP-OES; Avio 200, PerkinElmer, Waltham, MA, USA) was used to measure calcium and sodium ion content (*n* = 3). The dried CAG beads obtained after measuring the moisture content were collected and used as a dry sample for ICP-OES. The CAG bead preparation and drying process was repeated to collect approximately 0.3 g of dry sample for ICP-OES. Dry samples were completely dissolved in 2 mL ultrapure water, 4 mL nitric acid, and 0.5 mL hydrochloric acid using a microwave reaction system (Multiwave PRO, Anton Paar, Graz, Austria) and then ultrapure water was added to make 100 mL. The sodium ion content of the sample solution was measured at 589.592 nm by ICP-OES, while the calcium ion content was measured at 317.933 nm after the sample solution had been diluted 10-fold. Calibration curves were produced from 0 to 25 mg/L of calcium ions and 0 to 200 mg/L of sodium ions using standard solutions and were used to determine the calcium and sodium ion content. The ion and moisture contents of the dried CAG beads were used to calculate the ion content of the wet CAG beads.

### 2.8. Sodium Ions Diffusion of CAG Beads

CAG beads prepared at 5 °C were incubated with distilled water for 0, 30, and 60 min at room temperature (20 °C) to confirm whether the remaining sodium ions in the beads had diffused out. The CAG beads were then immediately frozen in liquid nitrogen, cut in half, and lyophilized. An energy-dispersive X-ray spectrometer (EDS; X-Max N, Oxford Instruments, Abingdon, UK) equipped with a field-emission scanning electron microscope (FE-SEM; MIRA 3, TESCAN, Brno, Czech Republic) was used to analyze the sodium ion content of the dried CAG beads at an accelerating voltage of 15 kV.

### 2.9. CAG Bead Microstructure

A low-vacuum scanning electron microscope (LV-SEM; JSM-6490LV, JEOL Ltd., Tokyo, Japan) was used to investigate the effect of gelation temperature on the microstructure of CAG beads. CAG beads prepared at each gelation temperature were frozen and dried using the method described in Section 2.8, coated with gold using an ion sputter, and observed by LV-SEM at an accelerating voltage of 15 kV (E-1010, Hitachi, Tokyo, Japan).

### 2.10. Density

To determine the density, the weight and diameter of CAG beads (*n* = 10) were analyzed. Weights were determined using a digital balance (Radwag AS 220-R1, Radom, Poland), using Equation (2) as follows:(2)$$\begin{array}{c} D=\frac{M}{V},\;V=\frac{4}{3}\;\pi r^{3}\\ D,\;\mathrm{density}\;(g/{\mathrm{cm}}^{3});\;M,\;\mathrm{weight}\;(g);\;V,\;\mathrm{volume}\;({\mathrm{cm}}^{3});\;r,\;\mathrm{radius}\;(\mathrm{cm}). \end{array}$$

## 3. Results and Discussion

### 3.1. Fitting the Models

The CCD matrix and experimental values of the dependent variables for each independent variable are presented in Table 2. The experimental values were used to calculate the regression coefficients of the constant, linear, quadratic, and interaction terms in the quadratic polynomial model equations for each dependent variable. Table 3 and Table 4 show the calculated coefficients and fitted equations, respectively. The constant and linear term coefficients for Y_1_ (diameter) and Y_3_ (rupture strength) were significant (*p* < 0.05), whereas the quadratic and interaction terms were not. The constant, X_1_, X_3_, X_1_X_1_, and X_3_X_3_ term coefficients for Y_2_ (sphericity) were significant (*p* < 0.05), thus implying a curvilinear effect of X_1_ and X_3_ on the sphericity of CAG beads, while all interaction terms for Y_2_ were not significant. The insignificant interaction terms for Y_1_, Y_2_, and Y_3_ mean that the effects of gelation temperature on the physical properties of CAG beads were not significantly related to the other factors—sodium alginate and calcium lactate concentration and gelation time. The determination coefficient (R^2^) of the fitted quadratic polynomial model equations for Y_1_, Y_2_, and Y_3_ were 0.913, 0.912, and 0.935, respectively, and the R^2^ values for all response surface models were highly significant (*p* < 0.01) [19]. Furthermore, the adjusted R-square (Adj R^2^) values for Y_1_, Y_2_, and Y_3_ were 0.811, 0.809, and 0.860, respectively. All R^2^ and Adj R^2^ values were greater than 0.8, indicating that the fitted equations adequately describe the effects of the independent variables on the diameter, sphericity, and rupture strength of CAG beads [20,21,22].

Analysis of variance (ANOVA) was used to evaluate the quality of the fitted response surface model equations [23]; the ANOVA results are shown in Table 5. The linear terms of all dependent variables were significant at the 99.9% probability level (*p* < 0.001) and the square term of Y_2_ was significant (*p* < 0.05). Conversely, the square terms of Y_1_ and Y_3_ and interaction terms of all dependent variables were insignificant (*p* > 0.05). The P-values for the lack-of-fit tests of all response surface models were higher than 0.05 (Y_1_, Y_2_, and Y_3_ were 0.415, 0.389, and 0.170, respectively), suggesting that the response surface models adequately explained the functional relationship between the dependent and independent variables [24].

### 3.2. Diameter and Sphericity

We used a three-dimensional response surface plot to visually display the effects of gelation temperature (X_3_) and other independent variables [sodium alginate (X_1_) and calcium lactate (X_2_) concentration, gelation time (X_4_)] on the physical properties of the CAG beads.

Figure 2a shows that the diameter (Y_1_) of the CAG beads increased with increasing sodium alginate concentration (X_1_) and decreased with increasing gelation temperature (X_3_). The effect of sodium alginate concentration and gelation temperature on the diameter of CAG beads might be explained by the viscosity of sodium alginate, which increased with increasing concentration or decreasing gelation temperature. As viscosity increased, the size of the sodium alginate droplets on the nozzle tip also increased, thereby increasing the diameter of the CAG beads [25,26]. Moreover, it can be seen from Figure 2a that the diameter (Y_1_) of the CAG beads decreased when the calcium lactate concentration (X_2_) or gelation time (X_4_) increased. These results can be explained by the study of Klokk et al. [26], in which the gel network was contracted by diffusing calcium ions into the sodium alginate droplets in the reactor.

Figure 2b shows that the sphericity (Y_2_) of the CAG beads increased as gelation temperature (X_3_) was slightly increased, and then decreased gradually. This is in contrast to the effect of sodium alginate concentration on the sphericity of CAG beads. CAG bead sphericity is closely related to sodium alginate viscosity; the shape of sodium alginate droplets is significantly altered when they hit the calcium lactate solution surface under low viscosity conditions, but sphericity is recovered by increasing the surface tension and gelation above a certain viscosity [9]. Consequently, the sphericity of the CAG beads gradually improved when sodium alginate viscosity increased (increasing sodium alginate concentration or decreasing gelation temperature); however, if the viscosity is too high, the falling sodium alginate droplets develop tails and the CAG beads eventually become tear-shaped [27]. An increase in the gelation time (X_4_) seemed to increase the sphericity of CAG beads slightly; however, the *P*-value of the X_4_ term coefficient for sphericity was greater than 0.05 (Table 3). These results suggest that gelation time does not have a significant effect on the sphericity of CAG beads likewise the calcium lactate concentration (X_2_). These results indicate that the shape of the falling sodium alginate droplets is an important factor determining the sphericity of CAG beads, and the influence of other factors is not significant after the formation of CAG beads formed through the reaction between alginate and calcium ions. In this study, CAG beads with sphericity greater than 95% were indistinguishable compared to perfect spheres when observed with the naked eye. Thus, preparation conditions must be carefully controlled to produce CAG beads with excellent visual sphericity.

### 3.3. Rupture Strength

Figure 2c shows that the rupture strength (Y_3_) of CAG beads increased proportionally with the sodium alginate concentration (X_1_), calcium lactate concentration (X_2_), and gelation time (X_4_). These results are consistent with previous studies that determined CAG gel strength according to the degree of interaction between calcium ions and α-L-guluronic acid, finding that strength was directly proportional to sodium alginate concentration, calcium concentration, and the duration of the interaction between alginate and calcium [28,29,30].

Gelation temperature is an important factor that significantly affects the rupture strength of CAG beads but has not been studied in detail to date. As Figure 2c indicates, the rupture strength (Y_3_) of the CAG beads increased when the gelation temperature (X_3_) decreased. Some studies have hypothesized that the increase in gel strength with low gelation temperature might be caused by the formation of a more dense internal structure due to a reduced calcium ion diffusion rate [31,32,33]; however, no experimental results or explanations yet support this theory. Consequently, we measured the calcium and sodium ion content of CAG beads prepared at different temperatures (5, 45, and 85 °C). Other factors were set as follows: sodium alginate concentration, 2.4%; calcium lactate concentration, 2.5%; gelation time, 18 min. As shown in Figure 3, the calcium ion content of the CAG beads decreased from 2.627 to 1.670 mg/g wet weight when the gelation temperature decreased from 85 to 5 °C. This means that the diffusion rate of calcium ions into sodium alginate droplets decreases with decreasing gelation temperature. For the same reasons, the sodium ion content of the CAG beads was highest (0.278 mg/g wet weight) when the gelation temperature was 5 °C.

Moreover, the sodium ion content of the CAG beads was highest (0.278 mg/g wet weight) when the gelation temperature was 5 °C. This might cause the CAG beads to swell because of ion-exchange between the residual sodium and calcium ions and can be a problem for their storage stability [34,35]. We assumed that sodium ions might remain in the core of CAG beads, produced at a low gelation temperature because of a reduced diffusion rate out of the beads. Thus, we analyzed the distribution of residual sodium ions in CAG beads prepared at 5 °C and immersed in distilled water for 0, 30, and 60 min. Figure 4 shows that residual sodium ions were detected in the core of the non-immersed CAG beads but not in CAG beads immersed in distilled water for 30 or 60 min. These results indicate that sodium ions remain in the core of the CAG beads because the rate of ion diffusion slows down at low gelation temperatures. Furthermore, immersion allows for sodium ion release, improving the storage stability of CAG beads prepared at low temperatures. The rupture strength of CAG beads generated at 5 °C was decreased upon immersion in distilled water (Table 6). Immersion for 30 min did not significantly alter the rupture strength. Accordingly, to improve the storage stability of CAG beads generated at low temperatures, the immersion time is required to be appropriately set.

### 3.4. Microstructure

Next, we investigated the effect of gelation temperature on the internal structure of CAG beads. To prevent excessive shrinkage during lyophilization and ensure easy cutting, the beads were lyophilized after being halved while frozen [10,36]. The CAG beads were prepared at a sodium alginate concentration, calcium lactate concentration, and gelation time of 2.4%, 2.5%, and 18 min, respectively. Figure 5 depicts the internal structure of CAG beads prepared at gelation temperatures of 5, 45, and 85 °C. At 30× magnification, the CAG beads prepared at 85 °C had a smooth and homogeneous microstructure, whereas the beads prepared at 5 °C had a rough microstructure containing some big cracks which were likely caused when the CAG beads were freeze-dried due to the relatively high moisture content and rupture strength (Figure 6). At 500× magnification, the CAG beads prepared at 5 °C displayed a microstructure with a well-connected bonding structure and a pattern, unlike those prepared at 45 and 85 °C. This regular microstructure is similar to the SEM image of CAG obtained by Topuz et al. [37]. This difference was better observed at 3000× magnification. Figure 7 shows that the density of CAG beads increased with an increase in the gelation temperature. The calcium ion diffusion rate increased with an increase in the gelation temperature; therefore, the CAG beads generated at a low gelation temperature had a more regular internal structure (not a more dense structure), thus conferring increased rupture strength.

### 3.5. Optimal Conditions for Maximum Rupture Strength

Lastly, we optimized the conditions to prepare CAG beads with maximum rupture strength (Y_3_). The optimal X_1_ (sodium alginate concentration), X_2_ (calcium lactate concentration), X_3_ (gelation temperature), and X_4_ (gelation time) conditions for preparing CAG beads with a maximum rupture strength were 3.6%, 4%, 5 °C, and 30 min, respectively (Table 7). Table 8 shows the percentage error verifying the accuracy of the predicted values under the optimal conditions. The predicted Y_1_ (diameter), Y_2_ (sphericity), and Y_3_ (rupture strength) values were 2.85 mm, 94.5%, and 6676 kPa, respectively. We prepared CAG beads under optimal conditions, yielding similar experimental Y_1_, Y_2_, and Y_3_ values of 2.88 ± 0.01 mm, 97.5 ± 0.9%, and 6444 ± 692 kPa, respectively. Consequently, the percentage error values (1.05, 3.17, and 3.48%, respectively) among the experimental and predicted values of Y_1_, Y_2_, and Y_3_ were very small, implying that the developed models were considerably fitted [38].

## 4. Conclusions

Here, we showed that gelation temperature is an important factor affecting the physical properties of CAG beads. Moreover, our study demonstrated that low gelation temperatures lowered the calcium ion diffusion rate, yielding a more regular microstructure to CAG beads. For this reason, the rupture strength of CAG beads increased as gelation temperature decreased. Furthermore, the CAG beads produced at a low gelation temperature might contain sodium ions; therefore, appropriate immersion is necessary to release these ions and improve the storage stability of CAG beads. Thus, we suggest that gelation temperature should be considered carefully in future research and development using CAGs.

## Figures and Tables

**Figure 1 foods-09-00180-f001:**
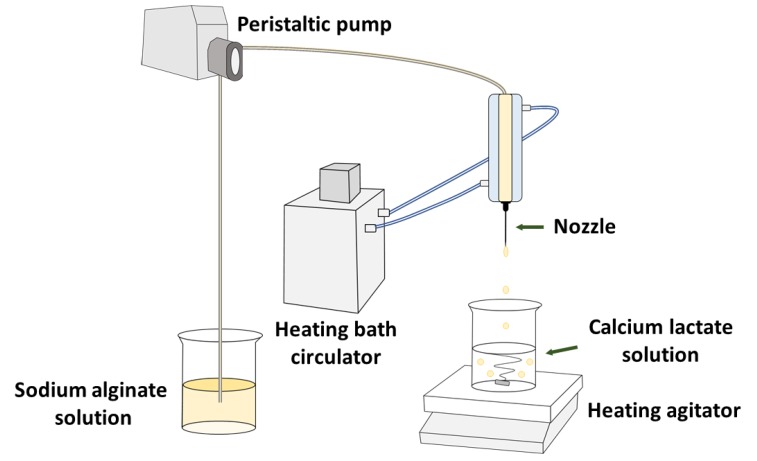
Simple schematic diagram of calcium alginate gel (CAG) bead preparation.

**Figure 2 foods-09-00180-f002:**
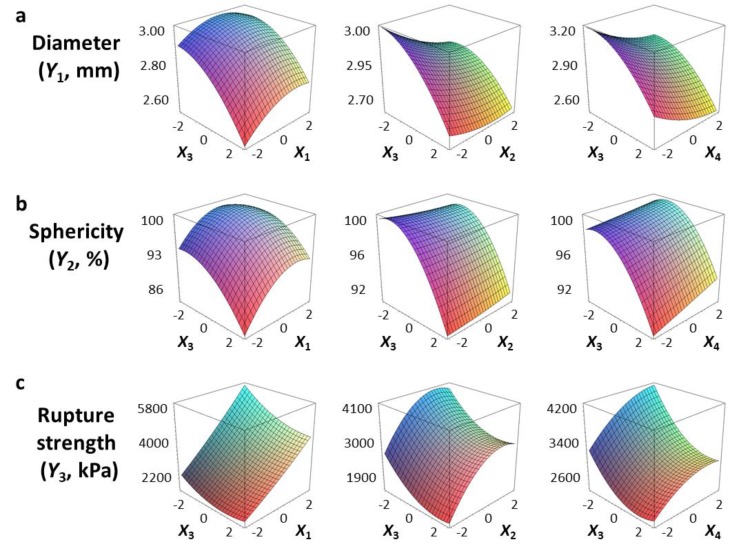
Three-dimensional response surface plots of the physical properties ((**a**), Diameter; (**b**), Sphericity; (**c**), Rupture strength) of CAG beads. X_1_, sodium alginate concentration (%, *w*/*v*); X_2_, calcium lactate concentration (%, *w*/*v*); X_3_, gelation temperature (°C); X_4_, gelation time (min).

**Figure 3 foods-09-00180-f003:**
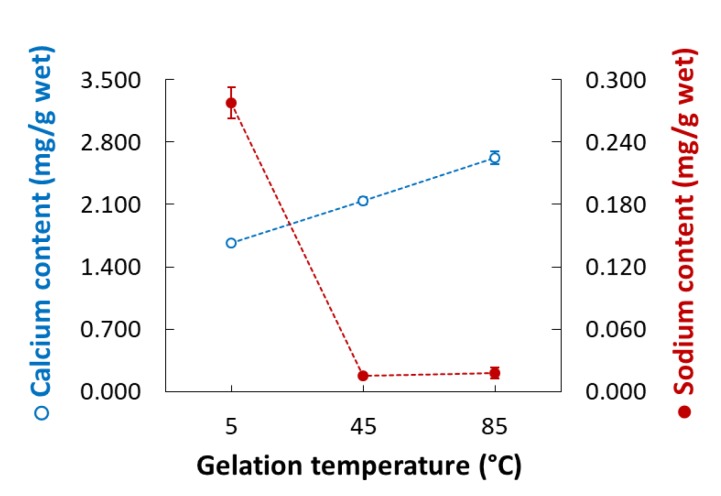
Calcium (open circles) and sodium ion (filled circles) content of CAG beads prepared at different gelation temperatures.

**Figure 4 foods-09-00180-f004:**
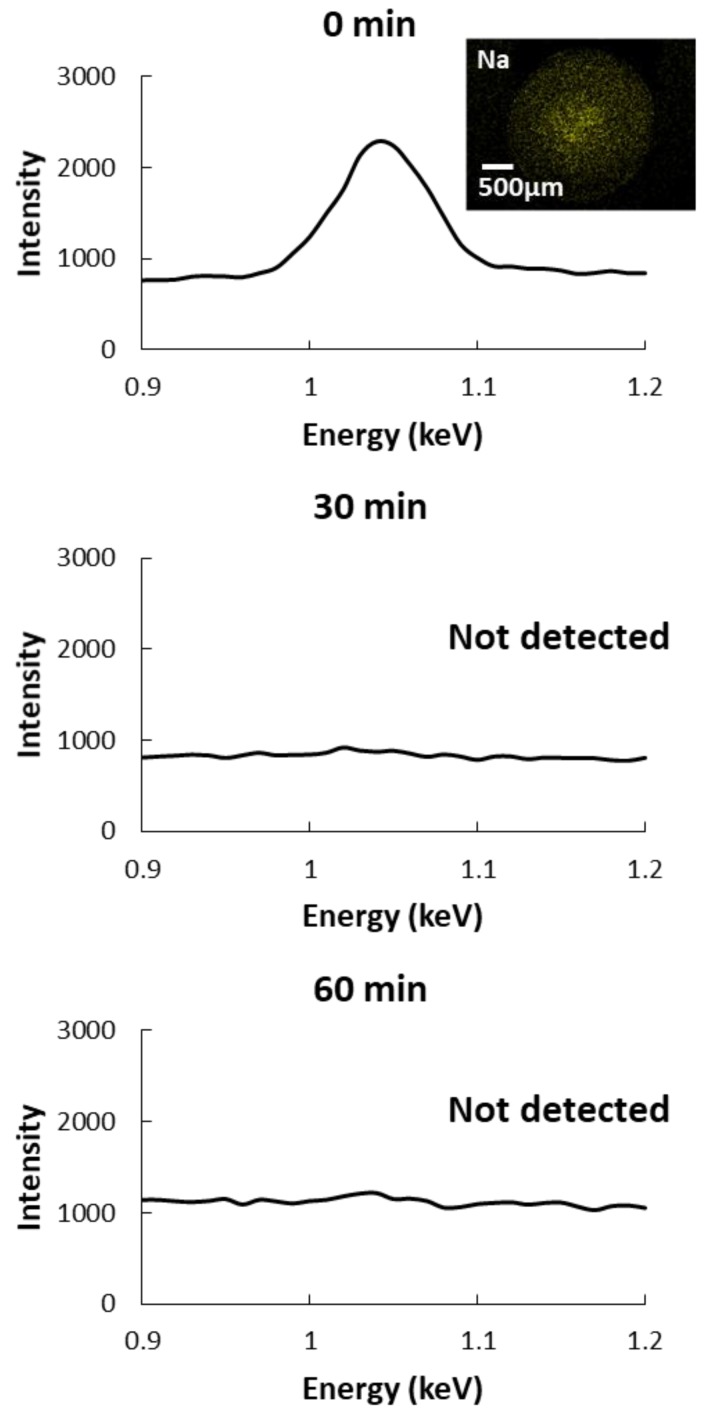
Energy-dispersive X-ray spectrometer (EDS) spectra and mapping results for sodium ions in CAG beads prepared at 5 °C after immersion in distilled water for different lengths of time.

**Figure 5 foods-09-00180-f005:**
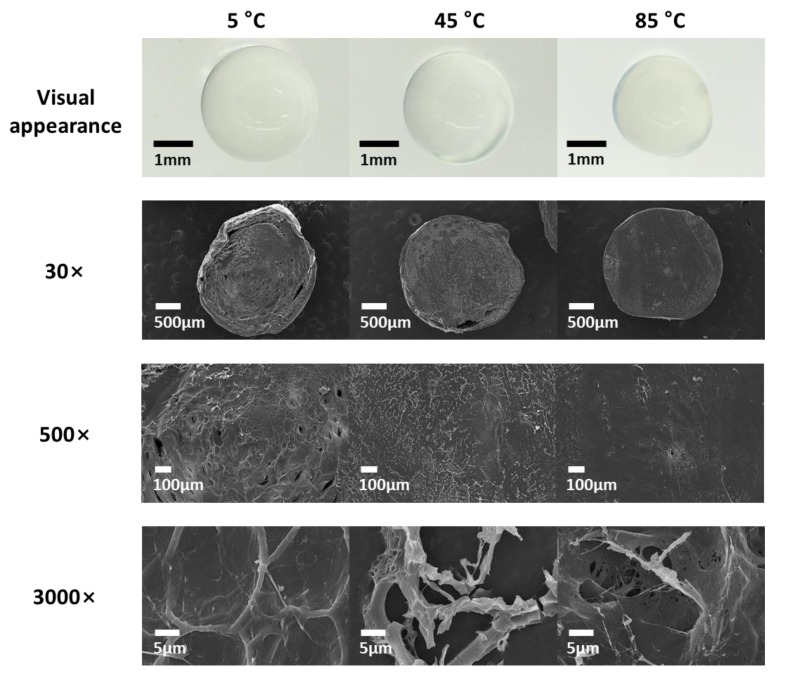
Visual appearance and low-vacuum scanning electron microscope (LV-SEM) images of CAG beads prepared at different gelation temperatures.

**Figure 6 foods-09-00180-f006:**
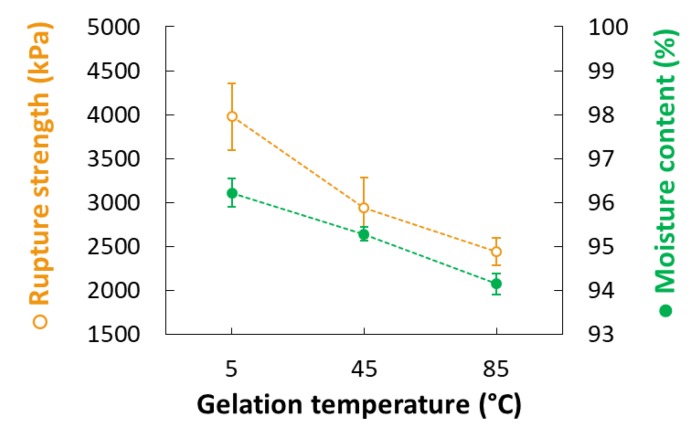
Rupture strength (open circles) and moisture content (filled circles) of CAG beads prepared at different gelation temperatures.

**Figure 7 foods-09-00180-f007:**
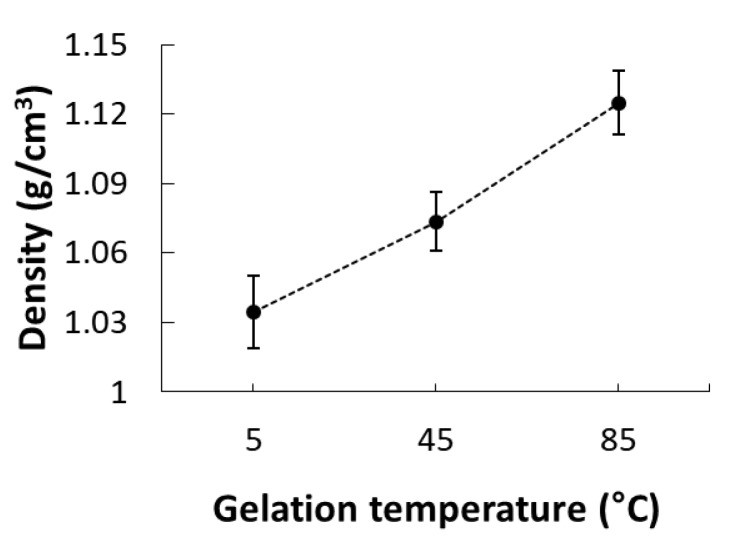
Density (g/cm^3^) of CAG beads prepared at different gelation temperatures.

**Table 1 foods-09-00180-t001:** The range and levels of the independent variables in central composite design (CCD) for monitoring the effects of preparation conditions on the physical properties.

Independent Variables	Symbol	Range and Levels				
		−2	1	0	1	2
Sodium alginate concentration (%, *w*/*v*)	X_1_	1.2	1.8	2.4	3.0	3.6
Calcium lactate concentration (%, *w*/*v*)	X_2_	0.5	1.5	2.5	3.5	4.5
Gelation temperature (°C)	X_3_	5	25	45	65	85
Gelation time (min)	X_4_	6	12	18	24	30

**Table 2 foods-09-00180-t002:** The CCD matrix and experimental values of the dependent variables for each independent variable.

Run No.		Independent Variables								Dependent Variables		
		Coded Values				Uncoded Values						
		X_1_	X_2_	X_3_	X_4_	X_1_	X_2_	X_3_	X_4_	Y_1_	Y_2_	Y_3_
Factorial	1	−1	−1	−1	−1	1.8	1.5	25	12	3.07	96.7	1993
portions	2	1	−1	−1	−1	3.0	1.5	25	12	3.08	98.9	3473
	3	−1	1	−1	−1	1.8	3.5	25	12	3.00	96.2	2274
	4	1	1	−1	−1	3.0	3.5	25	12	3.02	98.1	4005
	5	−1	−1	1	−1	1.8	1.5	65	12	2.82	92.1	1901
	6	1	−1	1	−1	3.0	1.5	65	12	2.88	95.4	2629
	7	−1	1	1	−1	1.8	3.5	65	12	2.81	91.6	2195
	8	1	1	1	−1	3.0	3.5	65	12	2.87	95.7	3606
	9	−1	−1	−1	1	1.8	1.5	25	24	2.93	97.8	2420
	10	1	−1	−1	1	3.0	1.5	25	24	2.99	99.2	3832
	11	−1	1	−1	1	1.8	3.5	25	24	2.91	98.3	2601
	12	1	1	−1	1	3.0	3.5	25	24	2.91	97.8	4500
	13	−1	−1	1	1	1.8	1.5	65	24	2.72	94.6	1959
	14	1	−1	1	1	3.0	1.5	65	24	2.77	95.5	3575
	15	−1	1	1	1	1.8	3.5	65	24	2.70	94.2	2087
	16	1	1	1	1	3.0	3.5	65	24	2.77	95.4	3902
Axial	17	−2	0	0	0	1.2	2.5	45	18	2.73	89.4	1436
portions	18	2	0	0	0	3.6	2.5	45	18	2.99	98.5	4420
	19	0	−2	0	0	2.4	0.5	45	18	3.14	96.6	1044
	20	0	2	0	0	2.4	4.5	45	18	2.82	98.1	3414
	21	0	0	−2	0	2.4	2.5	5	18	3.04	98.1	3976
	22	0	0	2	0	2.4	2.5	85	18	2.62	90.7	2440
	23	0	0	0	−2	2.4	2.5	45	6	3.09	96.7	2065
	24	0	0	0	2	2.4	2.5	45	30	2.88	97.8	3111
Center	25	0	0	0	0	2.4	2.5	45	18	2.97	98.3	2788
points	26	0	0	0	0	2.4	2.5	45	18	2.92	96.6	2942
	27	0	0	0	0	2.4	2.5	45	18	2.88	97.5	3110

X_1_: Sodium alginate concentration (%, *w*/*v*), X_2_: Calcium lactate concentration (%, *w*/*v*), X_3_: Gelation temperature (°C), X_4_: Gelation time (min). Y_1_: Diameter (mm), Y_2_: Sphericity (%), Y_3_: Rupture strength (kPa). Each experiment was carried out ten times and the mean value was used.

**Table 3 foods-09-00180-t003:** The regression coefficients of the fitted quadratic polynomial models for monitoring the effects of preparation conditions on the physical properties.

Parameter	Y_1_		Y_2_		Y_3_	
	Coefficient	*p*-Value	Coefficient	*p*-Value	Coefficient	*p*-Value
Constant	2.92333	0.001	97.4667	0.001	2946.67	0.001
X_1_	0.03542	0.011	1.3625	0.001	752.50	0.001
X_2_	−0.03792	0.007	0.0042	0.986	338.67	0.001
X_3_	−0.10042	0.001	−1.8042	0.001	−263.17	0.003
X_4_	−0.05292	0.001	0.4292	0.088	203.83	0.013
X_1_X_1_	−0.01969	0.141	−0.8198	0.006	28.04	0.713
X_2_X_2_	0.01031	0.426	0.0302	0.904	−146.71	0.072
X_3_X_3_	−0.02719	0.050	−0.7073	0.014	98.04	0.212
X_4_X_4_	0.01156	0.373	0.0052	0.983	−56.96	0.459
X_1_X_2_	−0.00188	0.899	−0.0688	0.812	101.25	0.262
X_1_X_3_	0.00937	0.528	0.2813	0.340	−59.50	0.502
X_1_X_4_	0.00187	0.899	−0.5313	0.085	87.00	0.331
X_2_X_3_	0.01188	0.427	0.0938	0.746	4.00	0.964
X_2_X_4_	0.00187	0.899	0.0063	0.983	−48.75	0.581
X_3_X_4_	0.00063	0.966	0.1063	0.714	−26.00	0.767

X_1_: Sodium alginate concentration (%, *w*/*v*), X_2_: Calcium lactate concentration (%, *w*/*v*), X_3_: Gelation temperature (°C), X_4_: Gelation time (min). Y_1_: Diameter (mm), Y_2_: Sphericity (%), Y_3_: Rupture strength (kPa).

**Table 4 foods-09-00180-t004:** The response surface model equations for monitoring the effects of preparation conditions on the physical properties.

Quadratic Polynomial Model Equations	R^2^	Adj R^2^	S	*p*-Value
Y_1_ = 2.92333 + 0.03542X_1_ − 0.03792X_2_ − 0.10042X_3_ − 0.05292X_4_ − 0.01969X_1_^2^ + 0.01031X_2_^2^ − 0.02719X_3_^2^ + 0.01156X_4_^2^ − 0.00188X_1_X_2_ + 0.00937X_1_X_3_ + 0.00187X_1_X_4_ + 0.01188X_2_X_3_ + 0.00187X_2_X_4_ + 0.00062X_3_X_4_	0.913	0.811	0.0577410	0.001
Y_2_ = 97.4667 + 1.3625X_1_ + 0.0042X_2_ − 1.8042X_3_ + 0.4292X_4_ − 0.8198X_1_^2^ + 0.0302X_2_^2^ − 0.7073X_3_^2^ + 0.0052X_4_^2^ − 0.0688X_1_X_2_ + 0.2813X_1_X_3_ − 0.5313X_1_X_4_ + 0.0938X_2_X_3_ + 0.0063X_2_X_4_ + 0.1063X_3_X_4_	0.912	0.809	1.13336	0.001
Y_3_ = 2946.67 + 752.50X_1_ + 338.67X_2_ − 263.17X_3_ + 203.83X_4_ + 28.04X_1_^2^ − 146.71X_2_^2^ + 98.04X_3_^2^ − 56.96X_4_^2^ + 101.25X_1_X_2_ − 59.50X_1_X_3_ + 87.00X_1_X_4_ + 4.00X_2_X_3_ − 48.75X_2_X_4_ − 26.00X_3_X_4_	0.935	0.860	343.729	0.001

X_1_: Sodium alginate concentration (%, *w*/*v*), X_2_: Calcium lactate concentration (%, *w*/*v*), X_3_: Gelation temperature (°C), X_4_: Gelation time (min). Y_1_: Diameter (mm), Y_2_: Sphericity (%), Y_3_: Rupture strength (kPa).

**Table 5 foods-09-00180-t005:** The analysis of variance (ANOVA) of response surface model equations for monitoring the effects of preparation conditions on the physical properties.

Dependent Variables	Sources	DF	SS	MS	*f*-Value	*p*-Value
Y_1_ Diameter (mm)	Regression					
	Linear	4	0.373817	0.093454	28.03	0.001
	Square	4	0.040404	0.010101	3.03	0.061
	Interaction	6	0.003838	0.000640	0.19	0.973
	Residual					
	Lack of fit	10	0.035942	0.003594	1.77	0.415
	Pure error	2	0.004067	0.002033	-	-
	Total	26	0.458067	-	-	-
Y_2_ Sphericity (%)	Regression					
	Linear	4	127.095	31.7738	24.74	0.001
	Square	4	25.677	6.4193	5.00	0.013
	Interaction	6	6.179	1.0298	0.80	0.587
	Residual					
	Lack of fit	10	13.967	1.3967	1.93	0.389
	Pure error	2	1.447	0.7233	-	-
	Total	26	174.365	-	-	-
Y_3_ Rupture strength (kPa)	Regression					
	Linear	4	19,002,146	4,750,537	40.21	0.001
	Square	4	1,093,213	273,303	2.31	0.117
	Interaction	6	390,870	65,145	0.55	0.760
	Residual					
	Lack of fit	10	1,365,918	136,592	5.27	0.170
	Pure error	2	51,875	25,937	-	-
	Total	26	21,904,022	-	-	-

DF (Degrees of freedom), SS (Sum of square), MS (Mean square).

**Table 6 foods-09-00180-t006:** Rupture strength of CAG beads generated at 5 °C after immersion in distilled water for different periods.

Immersion Time	0 min	30 min	60 min
Rupture strength	3910 ± 150 ^a^	3784 ± 119 ^a^	3187 ± 114 ^b^

^a,b^ The same letter indicates no significant difference (*p* < 0.05, Tukey’s range test).

**Table 7 foods-09-00180-t007:** The optimal conditions for preparing CAG beads with a maximum rupture strength.

Optimal Conditions			Y_3_ Rupture Strength (kPa)	
			Target Value	Maximum
X_1_ Sodium alginate concentration (%, *w*/*v*)	Coded value	2	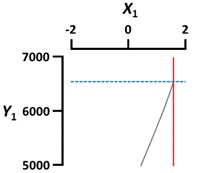	
	Actual value	3.6		
X_2_ Calcium lactate concentration (%, *w*/*v*)	Coded value	1.5	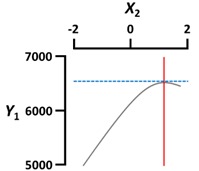	
	Actual value	4		
X_3_ Gelation temperature (°C)	Coded value	−2	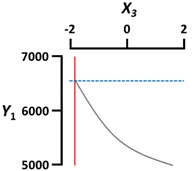	
	Actual value	4		
X_4_ Gelation time (min)	Coded value	2	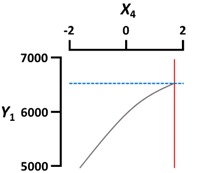	
	Actual value	30		

**Table 8 foods-09-00180-t008:** Verification of experimental and predicted values under optimal conditions.

	Y_1_ Diameter (mm)	Y_2_ Sphericity (%)	Y_3_ Rupture Strength (kPa)
Predicted values	2.85	94.5	6676
Experimental values	2.88 ± 0.01	97.5 ± 0.9	6444 ± 692
Error (%)	1.05	3.17	3.48

Optimal conditions: Sodium alginate concentration = 3.6%; Calcium lactate concentration = 4% min; Gelation temperature = 4 °C; Gelation time = 30 min. Error (%) = (Difference among predicted value and actual value/predicted value) × 100.

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
