# Peer review of "Changes in the Physical Properties of Calcium Alginate Gel Beads under a Wide Range of Gelation Temperature Conditions"

_foods, 2020, doi:10.3390/foods9020180_

Round 1

Reviewer 1 Report

The response surface methodology (RSM) was used to study the effect of several parameters on the properties of calcium alginate (CAG) beads by the authors.

A similar work was previously performed by the same authors (article that should be cited) on the influence of the heat treatment and time on the CAG beads (Seonghui Kim, Chungeun Jeong, Suengmok Cho and Seon-Bong Kim, Foods 2019, 8(11), 57; Effects of thermal treatment on physical properties of edible calcium alginate gel beads: Response surface methodological approach; received by Foods on the 15th of October 2019). Therefore, it would have been much more relevant to merge both studies in one paper to have a higher impact and an increased interest to the readers instead of dividing the study in two or more papers where the information and interest is significantly diluted. Unfortunately, this leads to a considerably reduced novelty and originality of the present paper. Major modifications are required.

In the introduction, a paragraph on the use and importance of CAG beads and their physical/mechanical properties related to the food industry should be given. Moreover, the use and relevance of RSM in the food industry should also be explained.

The abbreviation CCD (line 95) should be defined and the authors should discuss why they choose a central composite design matrix among all the designs.

In part 3.1. the authors recall that p values < 0.05 are significant meaning that the parameters have an influence on the properties of the CAG beads. They state that the interaction terms are not significative on the properties of the CAG beads however on the sphericity of the CAG beads some main interactions are non significative (like the calcium lactate concentration and the gelation time). The authors should further discuss this issue and link it to a chemical point of view by being critical.

In part 3.4., in the SEM images at 5 °C, all the CAG beads appear to have a porous structure (“and not a non-porous internal structure” as stated). It would have been of interest to quantify the porosity as well as the density of the obtained CAG beads to go further in the discussion.

In addition, what did the authors mean by: “an incomplete and pored microstructure” (line 269).

By comparing the predicted and experimental values at the optimal conditions for maximum rupture strength, the diameter and rupture strength values agreed well whereas it was not the case for the sphericity (predicted: 94.5% vs. experimental: 97.05 +/- 0.9 %) as claimed by the authors. Can the authors discuss this point and adapt it in the text.

In the conclusions, the authors wrote: “a denser and more patterned microstructure to control CAG beads.” This appears to be confusing/misleading with the results and discussion section where the authors wrote: “dense and non-porous internal structure”. Could the authors detail this point more in the results and discussion section.

Furthermore, results and a discussion on the influence of the immersion time of the CAG beads in water on the rupture strength of the beads should be added in the manuscript.

Finally, the authors should add a section where they discuss the influence of the thermal treatment (heating temperature and heating time) on the physical properties of the CAG beads at the optimal conditions.

Reviewer 2 Report

The article submitted is very interesting and the obtained results are useful especially in the industrial production of alginate beads for different purposes. However, several improvements need to be introduced into the text to improve its scientific quality.  They are presented below.

Page 2 Line 72-73. The Authors have established the experimental device on their own. If the temperature is affecting the sodium alginate solution why not lower the drop distance?

Page 3 line 91 why the rheometer was used not texturometer?

Page 3 Line 95. Describing the method please use the full name first then the abbreviation. Materials and methods chapters should be self-explanatory.

Page 4 Line 113. This is unclear - the moisture content of CAG beads was measured, or moisture content of 10 CAG beads was measured? What kind of measurement it was? Thermographic balance? What was the time of measurement?

Page 5 Line 156-158. The R2 value of an average 0.8 is not close to 1. please change this part. 

Page 7 Line 191. As the Authors are comparing the sphericity of CAG with fish roe it should be necessary to give some data about the average sphericity of fish roe to be able to compare.

Page 8 Line 14 It would be necessary to provide, as comparison value, any rupture strength values for fish roe.

Round 2

Reviewer 1 Report

The authors improved the clarity of the manuscript.

Some minor modifications in the text would need to be done (for example, l49, “CAGs has not (remove the yet”) been studied yet”; l86 All the prepared CAG beads” etc. and a small paragraph rewritten to make it clearer: from l231 to l235)
